# Towards Policy-Compliant Agents: Learning Efficient Guardrails for Policy Violation Detection

## Abstract

Autonomous web agents need to operate under externally imposed or human-specified policies while generating long-horizon trajectories. However, little work has examined whether these trajectories comply with such policies, or whether policy violations persist across different contexts such as domains (e.g., shopping or coding websites) and subdomains (e.g., product search and order management in shopping). To address this gap, we introduce POLICYGUARDBENCH, a benchmark of about 60k examples for detecting policy violations in agent trajectories. From diverse agent runs, we generate a broad set of policies and create both within subdomain and cross subdomain pairings with violation labels. In addition to full-trajectory evaluation, POLICYGUARDBENCH also includes a prefix-based violation detection task where models must anticipate policy violations from truncated trajectory prefixes rather than complete sequences. Using this dataset, we train POLICYGUARD-4B, a lightweight guardrail model that delivers strong detection accuracy across all tasks while keeping inference efficient. Notably, POLICYGUARD-4B generalizes across domains and preserves high accuracy on unseen settings. Together, POLICYGUARDBENCH and POLICYGUARD-4B provide the first comprehensive framework for studying policy compliance in web agent trajectories, and show that accurate and generalizable guardrails are feasible at small scales.

## 1 Introduction

Autonomous web agents are increasingly deployed to perform complex tasks in diverse environments, ranging from travel planning to automated transactions (Wang et al., 2024b; Abuelsaad et al., 2024; Qi et al., 2025). These agents often operate under externally imposed or human-specified policies, which are intended to constrain their behavior and ensure compliance with safety, regulatory and ethical requirements. As such agents generate long-horizon trajectories consisting of sequential actions, a central and underexplored question emerges: *to what extent do these trajectories adhere to the intended policies?*

Despite rapid advances in planning (Yao et al., 2023a; Zhou et al., 2024a; Singh et al., 2024), reasoning (Yao et al., 2023b; Shinn et al., 2023), and exploration (Shridhar et al., 2021; Wang et al., 2024a) of web agents, their mechanisms for policy compliance remains underexplored. Existing studies primarily focus on improving the capabilities of agents to complete tasks, yet rarely examines whether the actions taken along a trajectory satisfy explicit constraints. Moreover, compliance is not a local property of single steps. It depends on cumulative context, external rules, and domain or subdomain policies (Li & Waldo, 2024; Osogami, 2025). For example, purchasing alcohol may be allowed on a shopping website but prohibited in workplace procurement systems. Thus, a behavior permissible in one setting can breach rules in another, and violations often surface as actions are composed over long horizons. Without systematic detection and prevention of such violations, agents risk unintended and unsafe behaviors, which limits their reliability in practical deployments.

Studying policy-trajectory compliance presents several challenges. First, policies are inherently diverse and may originate from human instructions, institutional rules, or environmental constraints, which makes it non-trivial to align them with equally diverse trajectories. Second, long trajectories

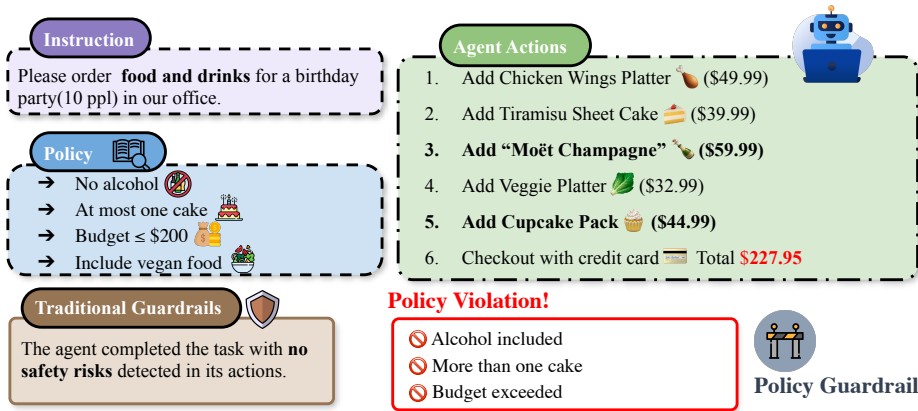

Figure 1: Example trajectory illustrating agent actions, policies, and violations. The agent completes the task but violates policies both directly (alcohol) and cumulatively (more than one cake, total cost > $200), cases that traditional guardrails fail to detect.

contain many actions, and it is unclear whether early prefixes carry enough signal to predict later violations, or whether full sequences are always necessary. Third, the absence of a large-scale and systematically annotated dataset prevents rigorous benchmarking of violation detection. Without such a resource, it is hard to measure progress or to design effective guardrails for web agents beyond narrow settings.

These challenges become particularly concrete in real-world deployment of web agents. As shown in Fig. 1, even a simple shopping task can lead to multiple policy violations: the agent may add alcohol despite explicit restrictions, include more than one cake, or exceed the specified budget when combining otherwise valid items. Such examples show that improving task completion alone is not enough—agents must also be evaluated and guided by their ability to remain *policy-compliant*. Similar combination-based violations also arise in other domains, such as scheduling multiple meetings that together exceed daily limits or making travel bookings that surpass company budgets. These violations are difficult to prevent with only hard-coded rules, underscoring the need for compliance-aware guardrails.

To address these challenges, we present **POLICYGUARDBENCH**, a 60k-scale benchmark for policy-trajectory violation detection. The benchmark is constructed by systematically deriving diverse policies from existing trajectories, and by forming both within-subdomain and cross-subdomain pairings. Each policy-trajectory pair is annotated for violation, which provides a reliable testbed for studying policy compliance. Beyond full-trajectory evaluation, POLICYGUARDBENCH also introduces a prefix-based violation detection task where models must predict violations from truncated trajectory prefixes. This task enables the study of early violation detection.

Building on this dataset, we further train **POLICYGUARD-4B**, a lightweight guardrail model that aims to balance efficiency and accuracy. Despite its small scale, POLICYGUARD-4B consistently outperforms strong open-source and closed-source baselines, and delivers robust performance across all tasks. Importantly, POLICYGUARD-4B also demonstrates cross-domain generalization, maintaining high accuracy even when evaluated on domains that are unseen during training.

In summary, our contributions are three-fold:

- We define **policy-trajectory compliance** as a core dimension of agent reliability, distinct from prior safety-oriented guardrails that focus on content filtering or single-step checks.

- We create **POLICYGUARDBENCH**, a 60k-scale benchmark built through a systematic pipeline for policy synthesis, trajectory matching, and violation annotation, with both cross-domain and prefix-based evaluation.

- We develop **POLICYGUARD-4B**, a lightweight and effective guardrail model that attains strong accuracy, cross-domain generalization, and state-of-the-art efficiency, demonstrating the practicality of small-scale guardrails in real-world deployment.

Together, these contributions establish a comprehensive foundation for the systematic study of policy compliance in web agent trajectories. Our results show that compact guardrails can be accurate and generalizable while remaining deployable at small scale, marking a concrete step toward trustworthy policy-compliant agents.

## 2 BACKGROUND: FROM SAFETY TO POLICY COMPLIANCE

Current research on guardrails for language agents has largely emphasized *safety-oriented* objectives (Inan et al., 2023; Lee et al., 2025; Liu et al., 2025; Wen et al., 2025; Ghosh et al., 2025), such as detecting harmful prompts, preventing unsafe completions, or enforcing reliability constraints. These efforts are well motivated, since safety is a prerequisite for deployment, but they often result in over-defensiveness: models may over-refuse benign behaviors in order to minimize risks, thereby reducing usability in practice. A subset of recent work has begun to explicitly consider guardrails for autonomous agents, yet the focus remains primarily on catastrophic safety risks.

While prior work has focused primarily on safety, recent benchmarks show that web agents frequently violate policy constraints, with ST-WebAgentBench (Levy et al., 2024) reporting a significant gap between task completion and Completion-Under-Policy (CuP), and SafeArena (Tur et al., 2025) and WebSuite (Li & Waldo, 2024) likewise finding that agents often break policy requirements. Notably, some prior works have treated policy violations as equivalent to unsafe behaviors (Chen et al., 2025a;b; Vijayvargiya et al., 2025), an assumption that we argue is unfounded. Our own tests further corroborate this view: safety-oriented guardrails such as LLAMAGUARD3 (Meta Llama Team, 2024) and LLAMAGUARD4 (Meta Llama Team, 2025) often classify trajectories that clearly violate policies as *safe*. This not only highlights the conceptual distinction between safety and compliance, but also reveals a second issue: current guardrail methods fail to generalize to the task of detecting policy-trajectory violations. We provide a more detailed empirical analysis of these observations in §4.

Against this backdrop, recent work on agent guardrails can roughly be grouped into three directions. The first direction centers on adaptive detection of safety risks. For example, AGRAIL (Luo et al., 2025) is a lifelong guardrail that dynamically generates task-specific safety checks and adapts them at test time, introducing the Safe-OS benchmark to capture attacks against OS-level agents. A second line of work emphasizes formal policy verification. SHIELDAGENT (Chen et al., 2025b) constructs verifiable safety policies by translating natural language rules into probabilistic circuits, enabling trajectory-level checking of whether an agent's behavior satisfies formalized constraints. A third line pursues system-level and benchmark-driven approaches. LLAMAFIREWALL (Chennabasappa et al., 2025) integrates modular defenses-PromptGuard for jailbreak detection, AlignmentCheck for reasoning misalignment, and CodeShield for insecure code generation—and has been deployed in production as a layered defense system. Meanwhile, benchmark-driven efforts provide large-scale evaluations of agent behaviors, focusing on unsafe actions and high-risk outcomes across diverse environments (Zhou et al., 2024c; Vijayvargiya et al., 2025; Zheng et al., 2025; Andriushchenko et al., 2025). These datasets are valuable for assessing safety vulnerabilities, but they do not explicitly address policy–trajectory compliance.

Together, these efforts reveal a diverse landscape: some works adaptively detect risks, others enforce policies through formal verification, and still others focus on system-level security or large-scale safety evaluation. Yet a common theme is their emphasis on safety and catastrophic risksr, such as injection attacks, data leaks, or insecure code execution. Much less attention has been paid to policy compliance: whether an agent's trajectory conforms to task-specific, domain-specific, or externally imposed rules. This gap motivates our introduction of POLICYGUARDBENCH, which explicitly targets the detection of policy-trajectory violations, providing a large-scale benchmark and lightweight guardrail model to fill this missing dimension.

## 3 CONSTRUCTING POLICYGUARDBENCH AND POLICYGUARD-4B

We now describe the construction of POLICYGUARDBENCH and the training of our guardrail model POLICYGUARD-4B. POLICYGUARDBENCH is a 60k-scale dataset of trajectory-policy pairs annotated for violation, supporting both full-trajectory and prefix-based detection tasks across five domains. Based on this benchmark, we train POLICYGUARD-4B, a 4B-parameter lightweight

guardrail model. Below, we detail the dataset construction pipeline, data splits, and model training procedure.

## 3.1 DATA CONSTRUCTION

**Trajectory Standardization.** We start from raw trajectories produced by the SCRIBEAGENT (Shen et al., 2024) in the WEBARENA (Zhou et al., 2024b). The raw logs record heterogeneous browser events such as clicks, form inputs, scrolls, URL navigations, and status updates. To transform these into a clean and uniform representation, we remove noise (e.g., empty or duplicate events, rendering artifacts), normalize actions into a controlled verb set (Click, Input, Scroll, Select, Navigate, Submit, etc.), and canonicalize objects (e.g., link 'My Account' or button 'Search'). Redundant operations are merged, and excessively long trajectories may be truncated. Each trajectory is then assigned a domain and subdomain, which are propagated to all steps. Domains correspond to broad platforms such as *shopping* or *gitlab*, while subdomains capture finer-grained contexts within them (e.g., *products* in shopping, *help* in gitlab). The final output is serialized into a step-wise textual form, such as Step 1: Click link 'My Account'; Step 2: Scroll page; Step 3: Click button 'View Order'. In total, this process yields 733 standardized trajectories, which serve as the basis for downstream policy synthesis, matching, and annotation.

**Policy Synthesis.** To capture compliance constraints, we synthesize 2–3 candidate policies for each trajectory. These policies are designed to capture compliance requirements in a precise and checkable form, following principles of atomicity (one constraint per rule), executability, and clarity. Examples include *"Must confirm user intent before clicking on 'My Account' or 'My Orders' links"*, *"Do not submit a form without filling mandatory fields"*, and *"Do not click 'Delete' without a prior confirmation step"*. Candidate policies are first synthesized by GPT-4o[1] and subsequently curated to ensure quality and consistency.

Each policy is assigned a structured schema consisting of a source_subdomain and up to two additional target_subdomains within the same domain, enabling both within- and cross-subdomain evaluation. Policies are normalized for formatting, deduplicated semantically, and filtered to remove overly broad or unverifiable cases. Altogether, we obtain 2,195 curated policies, providing a diverse rule set for subsequent trajectory matching and violation annotation.

**Trajectory–Policy Matching.** For each domain, we construct trajectory–policy pairs through a two-stage matching process. First, candidate policies are retrieved for each trajectory using embedding-based similarity (Reimers & Gurevych, 2019) and keyword triggers (e.g., element names such as confirm or delete). These candidates are further refined using heuristic rules and LLM-based scoring to ensure that the policies are semantically relevant and checkable. [2]

Each policy is paired not only with trajectories from its source subdomain—the subdomain in which the policy was originally induced—but also with trajectories from up to two additional target subdomains within the same domain, enabling systematic evaluation of both within- and cross-subdomain generalization. Pairs are labeled by their type (source_subdomain vs. target_subdomain), yielding both within- and cross-subdomain samples. Since each trajectory typically generates 2-3 policies, this expansion produces a large number of pairs while remaining restricted to the same domain; detailed statistics are provided in Tab. 1.

**Violation Annotation.** Each trajectory–policy pair is labeled as either violation or no_violation. We define operational criteria covering common rule types: obligations (a required action is missing), prohibitions (a forbidden action is present), ordering constraints (steps occur in the wrong sequence), and conditional rules (a consequent is not satisfied when the antecedent holds). These criteria allow consistent and checkable mappings from trajectories to labels.

---

[1]We prompt GPT-4o (OpenAI, 2024) with a trajectory and its outcome to produce candidate rules, followed by manual filtering and refinement.

[2]To balance the dataset, we also add negative samples by pairing each trajectory with randomly chosen policies from the same domain that it does not violate, while ensuring that these pairings do not accidentally create false violations.

| Domain | Full Construction | | | POLICYGUARDBENCH | | |
|--------|--------|--------|--------|--------|--------|--------|
| | **Source** | **Target** | **Total** | **Source** | **Target** | **Total** |
| Reddit | 33,708 | 0 | 33,708 | 5,022 | 0 | 5,022 |
| Map | 49,152 | 0 | 49,152 | 5,005 | 0 | 5,005 |
| GitLab | 78,063 | 9,250 | 87,313 | 2,624 | 2,624 | 5,248 |
| Shopping_Admin | 33,025 | 29,648 | 62,673 | 8,347 | 8,347 | 16,694 |
| Shopping | 31,763 | 49,947 | 81,710 | 14,014 | 14,014 | 28,028 |
| **Total** | 225,711 | 88,845 | 314,556 | 35,012 | 24,985 | 59,997 |

Table 1: Statistics of the constructed Policy-Trajectory datasets. The *full construction* yields 2,195 policies, 733 trajectories, and over 314K trajectory-policy pairs, of which 28.2% involve cross-subdomain generalization (target pairs). Reddit and Map are source-only because their trajectories are confined to a single subdomain. From this pool, we curate a balanced 59,997-pair subset, denoted as POLICYGUARDBENCH, used for training and evaluation.

Annotation is carried out in two stages. In the first stage, we sample trajectories across different domains and manually annotate a subset, establishing consistent labeling guidelines. In the second stage, we employ gpt-oss-120B (OpenAI, 2025) to imitate the annotation patterns of human labelers, producing both labels and confidence scores. Cases with low confidence or inconsistent predictions are flagged for additional human review. This procedure yields consistent violation/no-violation annotations at scale.

## 3.2 DATA SPLITS

The full construction process yields over 314K trajectory–policy pairs, but this raw scale is not directly suitable for evaluation: the distribution is highly imbalanced, with domains such as Reddit and Map contributing only source-subdomain examples, while others dominate with far larger pair counts. To create a more balanced and tractable benchmark, we curate a 59,997-pair subset that equalizes both labels and generalization settings. This balanced dataset contains 25,435 violation cases (42.4%) and 34,562 non-violation cases, with 35,012 source pairs and 24,985 target pairs (41.6% cross-subdomain). We split the data into training and test sets with an 8:2 ratio, resulting in 49,997 training pairs and 12,000 test pairs. Coverage across all five domains is preserved, with detailed statistics provided in Tab. 1.

In addition to full-trajectory evaluation, we construct prefix-based splits to probe model robustness under partial information. Standardized trajectories contain an average of 9.3 actions; for violation cases, we truncate trajectories to the first $N$ steps ($N = 1 \ldots 5$), re-match them with their corresponding policies, and re-label the resulting pairs. This setting is motivated by the study of early decision-making under incomplete observations (Kumar et al., 2025; Bian et al., 2025; Otth et al., 2025), enabling us to assess whether models can detect early signals (e.g. partial action sequences suggesting upcoming violations) of policy violations before observing the full trajectory.

## 3.3 TRAINING POLICYGUARD-4B

We train POLICYGUARD-4B by instruction-tuning a Qwen3-4B-Instruct (Qwen Team, 2025a) backbone using full-parameter supervised fine-tuning. Training data is drawn from POLICYGUARD-BENCH, where each input follows a unified template concatenating the policy, the trajectory actions, and domain metadata, while the output is a binary label (violation/no_violation) under strict instruction formatting. This formulation casts policy–trajectory compliance detection as a single-task instruction-following problem. Optimization details such as learning rate, batch size, and number of epochs are reported in Appx. §A.

## 4 EVALUATING POLICY COMPLIANCE DETECTION

We evaluate POLICYGUARD-4B on POLICYGUARDBENCH through a series of experiments designed to assess both accuracy and efficiency. Our analysis spans standard benchmark comparisons, prefix-based violation detection, cross-domain generalization, and inference efficiency, providing a comprehensive picture of POLICYGUARD-4B's performance as a lightweight guardrail.

| Model | Type | Size | Accuracy | F1 | Latency |
|---|---|---|---|---|---|
| *Frontier Models (API tested)* | | | | | |
| DeepSeek-V3.1 (Non-thinking Mode) | Open | 685B | 0.8613 | 0.8407 | 3270.0 (1072.1%) |
| Gemini-1.5-Pro | Closed | – | 0.8713 | 0.8502 | 596.1 (195.4%) |
| Claude-Sonnet-4 | Closed | – | 0.8983 | 0.8678 | 1238.0 (405.9%) |
| *Open-source Foundation Models (Llama family)* | | | | | |
| Llama-3.2-3B-Instruct | IT | 3B | 0.6067 | 0.2767 | 44.3 (14.5%) |
| Llama-3.1-8B-Instruct | IT | 8B | 0.6647 | 0.4222 | 85.0 (27.9%) |
| Llama-3.3-70B-Instruct | IT | 70B | **0.9054** | **0.8883** | 305.0 (100.0%) |
| Llama-4-Scout-17B-16E-Instruct | IT | 109B | 0.8457 | 0.8198 | 265.0 (86.9%) |
| *Open-source Foundation Models (Qwen family)* | | | | | |
| Qwen3-4B-Instruct | IT | 4B | 0.6897 | 0.5348 | 25.6 (8.4%) |
| Qwen3-8B | PT | 8B | 0.6408 | 0.6407 | 115.8 (38.0%) |
| Qwen3-30B-A3B-Instruct | IT | 31B | 0.6183 | 0.6720 | 250.0 (82.0%) |
| Qwen2.5-72B-Instruct | IT | 72B | 0.8825 | 0.8607 | 205.0 (67.2%) |
| Qwen3-235B-A22B-Instruct-2507 | IT | 235B | 0.8869 | 0.8690 | 3640.0 (1193.4%) |
| *Open-source Foundation Models (Gemma family)* | | | | | |
| Gemma-3-4B | IT | 4B | 0.7876 | 0.7764 | 70.8 (23.2%) |
| Gemma-3-12B | IT | 12B | 0.8964 | 0.8773 | 51.3 (16.8%) |
| Gemma-3-27B | IT | 27B | 0.8850 | 0.8520 | 73.6 (24.1%) |
| *Safety Guardrail Models* | | | | | |
| Llama Guard-7B | Guardrail | 7B | 0.4256 | 0.5957 | 87.5 (28.7%) |
| Llama Guard-2-8B | Guardrail | 8B | 0.5753 | 0.0016 | 40.0 (13.1%) |
| Llama Guard-3-8B | Guardrail | 8B | 0.4246 | 0.5952 | 164.8 (54.0%) |
| Llama Guard-4-12B | Guardrail | 12B | 0.4239 | 0.5954 | 175.3 (57.5%) |
| ShieldGemma-2B | Guardrail | 2B | 0.5735 | 0.3317 | 32.6 (10.7%) |
| ShieldGemma-9B | Guardrail | 9B | 0.5457 | 0.3472 | 39.8 (13.0%) |
| ShieldGemma-27B | Guardrail | 27B | 0.5555 | 0.1834 | 45.0 (14.8%) |
| *Ours* | | | | | |
| POLICYGUARD-4B | FT | 4B | 0.9014 | 0.8759 | **22.5 (7.4%)** |

Table 2: Benchmark performance on POLICYGUARDBENCH. We report Accuracy, F1, and inference latency (ms/example) across open-source foundation models (Qwen, Llama, Gemma), safety guardrails (Llama Guard, ShieldGemma), and frontier models (DeepSeek, Gemini, Claude). IT = instruction-tuned, PT = pretrained, FT = Finetuned. Our fine-tuned POLICYGUARD-4B achieves strong accuracy while maintaining substantially lower latency compared to larger baselines.

## 4.1 EVALUATION SETUPS

**Evaluation protocol.** All experiments are conducted on POLICYGUARDBENCH, using the balanced 59,997-pair subset (See details in §3.2). The task is formulated as binary classification (violation/no_violation), so we report both Accuracy and F1 as the primary evaluation metrics. To additionally account for the practical requirement that guardrails be lightweight and efficient, we measure inference latency in milliseconds per example, enabling a fair comparison of accuracy–efficiency trade-offs across models.

**Baselines.** We compare POLICYGUARD-4B against three categories of baselines: (1) open-source foundation models including the Qwen family (Qwen Team, 2025b), Llama family (Meta AI, 2025) and Gemma family (Gemma Team, 2025); (2) safety-oriented guardrails, specifically the LlamaGuard family and ShieldGemma family (Zeng et al., 2024); and (3) frontier systems such as DeepSeek-V3.1(Non-thinking Mode) (Deepseel-AI, 2024), Gemini-1.5-Pro (Gemini Team, 2024) and Claude-Sonnet-4 (Anthropic, 2025) which we adapt through prompting to perform binary policy–trajectory classification[3]. All experiments are run on H100 80GB GPUs with deterministic decoding (temperature set to 0).

---

[3]Gemini-2.5-Pro automatically blocks the outputs of our task queries, so we instead report results using Gemini-1.5-Pro. Ethical and safety implications of this restriction will be discussed in the discussion section.

| Model | N=1 | N=2 | N=3 | N=4 | N=5 | Avg. |
|---|---|---|---|---|---|---|
| *Frontier Models* | | | | | | |
| DeepSeek-V3.1 (Non-thinking Mode) | 0.9271 | 0.8587 | 0.8467 | 0.8304 | 0.8129 | 0.8552 |
| Gemini-1.5-Pro | 0.8990 | 0.8779 | 0.8667 | 0.8630 | 0.8543 | **0.8722** |
| Claude-Sonnet-4 | 0.8994 | 0.8537 | 0.8484 | 0.8309 | 0.8115 | 0.8488 |
| *Open-source Foundation Models (Llama family)* | | | | | | |
| Llama-3.2-3B-Instruct | 0.9086 | 0.8199 | 0.7348 | 0.6377 | 0.5693 | 0.7341 |
| Llama-3.1-8B-Instruct | 0.8976 | 0.7741 | 0.6731 | 0.6121 | 0.5371 | 0.6988 |
| Llama-3.3-70B-Instruct | 0.9298 | 0.8441 | 0.8368 | 0.8305 | 0.8191 | 0.8521 |
| Llama-4-Scout-17B-16E-Instruct | 0.9389 | 0.8854 | 0.8583 | 0.8355 | 0.8237 | 0.8684 |
| *Open-source Foundation Models (Qwen family)* | | | | | | |
| Qwen3-4B-Instruct | 0.8832 | 0.8231 | 0.8038 | 0.7688 | 0.7330 | 0.8024 |
| Qwen3-8B | 0.6102 | 0.7429 | 0.6308 | 0.5526 | 0.4911 | 0.6055 |
| Qwen3-30B-A3B-Instruct | 0.7199 | 0.6955 | 0.7213 | 0.7273 | 0.7468 | 0.7222 |
| Qwen2.5-72B-Instruct | 0.8508 | 0.8170 | 0.8382 | 0.8404 | 0.8327 | 0.8358 |
| Qwen3-235B-A22B-Instruct-2507 | 0.8976 | 0.8752 | 0.8644 | 0.8569 | 0.8498 | 0.8688 |
| *Open-source Foundation Models (Gemma family)* | | | | | | |
| Gemma-3-4B-IT | 0.8260 | 0.6401 | 0.5781 | 0.6236 | 0.6385 | 0.6613 |
| Gemma-3-12B-IT | 0.9227 | 0.8492 | 0.8364 | 0.8258 | 0.8203 | 0.8509 |
| Gemma-3-27B-IT | 0.9108 | 0.8789 | 0.8526 | 0.8254 | 0.8099 | 0.8555 |
| *Ours* | | | | | | |
| POLICYGUARD-4B | 0.9101 | 0.8648 | 0.8441 | 0.8276 | 0.8190 | 0.8531 |

Table 3: Prefix-based violation detection accuracy on POLICYGUARDBENCH. We report performance at different prefix lengths $N = 1 \ldots 5$ and their average. Frontier models and large open-source LLMs show strong performance but at high inference cost, whereas our lightweight POLICYGUARD-4B achieves competitive accuracy across all prefix settings.

## 4.2 BENCHMARK PERFORMANCE

Overall, results on POLICYGUARDBENCH in Tab. 2 reveal three consistent trends: (1) large foundation models achieve strong accuracy but incur heavy inference costs, (2) existing safety-oriented guardrails fail to transfer to policy compliance detection, and (3) our lightweight POLICYGUARD-4B strikes the best balance of accuracy and efficiency.

**Capacity–Efficiency Trade-off.** General-purpose foundation models such as the Qwen and Llama families exhibit a familiar scaling pattern. Larger variants (e.g., Qwen2.5-72B, Llama-3.3-70B) achieve accuracies above 88% but require substantial latency (200–360 ms/example), while smaller 3–8B models reduce inference time but drop to the 61–69% range. This highlights a trade-off between capacity and deployability, complicating the use of such models as real-time guardrails.

**Mismatch of Safety Guardrails.** In contrast, safety-oriented guardrails such as the LlamaGuard family and ShieldGemma perform poorly on our benchmark. LlamaGuard outputs are highly skewed, often labeling nearly all inputs as either *safe* or *unsafe*, which reflects their coarse-grained training objectives. As a result, accuracy remains mostly below 60% and precision/recall are ill-defined, preventing meaningful F1 evaluation. ShieldGemma adds a classification head but still yields only mediocre performance, showing that safety supervision does not translate into policy compliance detection. These findings confirm that safety detection and policy compliance represent orthogonal dimensions of agent reliability, underscoring the necessity of a dedicated framework.

**Effectiveness of POLICYGUARD** Finally, our POLICYGUARD-4B reaches 90.1% accuracy with 22.5 ms/example latency (87.6% F1), surpassing substantially larger open- and closed-source baselines while remaining efficient. This demonstrates that guardrails explicitly trained for policy–trajectory compliance can be both accurate and deployable in practice.

### 4.3 CASE STUDY I: PREFIX-BASED VIOLATION DETECTION

A natural question is whether policy violations can be anticipated from only partial information rather than full trajectories. To probe this, we truncate each input to the first $N$ steps ($N = 1 \ldots 5$) and evaluate models under these prefix-based settings. As shown in Fig. 3, performance is generally highest at $N = 1$ and decreases as prefixes become longer, highlighting the inherent difficulty of violation detection under partial observations. Detailed numerical results are reported in Tab. 3, and more fine-grained visualization of model group trends is provided in Appx. §C.

**Scale drives robustness.** Within each family, larger models consistently maintain higher accuracy under truncation, while smaller variants degrade sharply. For example, Llama-4-Scout-17B-16E and Qwen3-235B remain robust even with $N = 1$, whereas Llama-3.2-3B and Qwen3-8B collapse as the prefix length grows. Similarly, frontier models such as DeepSeek-V3.1 and Gemini-1.5-Pro outperform most open-source baselines across all prefix settings, reinforcing the advantage of scale in capturing early predictive signals.

**Effects of truncation.** Across nearly all models, performance decreases monotonically as more actions are included. While shorter prefixes are more often compliant—making classification at small N partly easier by prior—certain violations do manifest early (e.g., direct access to restricted content), and effective models must be sensitive to such rare early signals. At the same time, longer prefixes introduce both more opportunities for violations and more contextual noise, which together make detection increasingly difficult.

**Policy-specific guardrails excel under partial information.** Despite being far smaller than frontier or 70B+ open-source models, our fine-tuned POLICYGUARD-4B achieves an average of 85.3% accuracy across all prefix settings. This demonstrates that targeted training for policy–trajectory compliance not only generalizes to full-trajectory evaluation but also remains effective in early detection scenarios, where robustness under partial observation is critical.

### 4.4 CASE STUDY II: LEAVE-ONE-DOMAIN-OUT GENERALIZATION

In practical deployments, web agents frequently encounter domain shifts, raising the question of whether policy–trajectory guardrails can generalize beyond the environments they are trained on. To investigate this, we adopt a leave-one-domain-out (LODO) protocol: in each split, one domain is held out for out-of-domain (OOD) testing, while the remaining four domains are used for training and in-domain (ID) evaluation, as reported in Tab. 4.

**Stable in-domain learning.** Across all five splits, in-domain (ID) performance remains stable, with both accuracy and F1 around 93%. This indicates that policy–trajectory compliance patterns are learned robustly within training domains, without overfitting to subdomain-specific artifacts.

| Domain | ID | | OOD | |
|---|---|---|---|---|
| | Accuracy | F1 | Accuracy | F1 |
| GitLab | 0.9314 | 0.9272 | 0.9116 | 0.9116 |
| Map | 0.9361 | 0.9343 | 0.9020 | 0.9078 |
| Reddit | 0.9326 | 0.9338 | 0.9024 | 0.9055 |
| Shopping | **0.9362** | **0.9370** | **0.9174** | **0.9137** |
| Shopping-Admin | 0.9276 | 0.9288 | 0.9079 | 0.9044 |
| Average | 0.9328 | 0.9322 | 0.9083 | 0.9086 |

Table 4: Leave-one-domain-out (LODO) results on POLICY-GUARDBENCH. We report Accuracy and F1 for both in-domain (ID) and out-of-domain (OOD) across five domains.

**Robust out-of-domain transfer.** When transferred to unseen domains, performance drops only moderately (average 90.8% accuracy and 90.9% F1).

The small gap of about 2–3 percentage points suggests that the model captures transferable compliance regularities rather than overfitting to domain-specific artifacts. Notably, Shopping yields the strongest OOD performance (91.7% Acc / 91.4% F1), whereas Map and Reddit show slightly larger gaps, reflecting their more heterogeneous action structures.

| Model | Size($\downarrow$) | F1 ($\uparrow$) | Latency($\downarrow$) | FLOPs ($\downarrow$) | EA-F1 ($\uparrow$) |
|---|---|---|---|---|---|
| *Frontier Models* | | | | | |
| DeepSeek-V3.1 (Non-thinking) | 685B | 0.8407 | 3270.0 | – | 0.2571 |
| Gemini-1.5-Pro | – | 0.8502 | 596.1 | – | 1.4263 |
| Claude-Sonnet-4 | – | 0.8678 | 1238.0 | – | 0.7010 |
| *Open-source Foundation Models* | | | | | |
| Gemma-3-12B-IT | 12B | 0.8773 | 51.3 | 7.87 | 17.1014 |
| Gemma-3-27B-IT | 27B | 0.8520 | 73.6 | 17.99 | 11.5761 |
| Llama-3.3-70B-Instruct | 70B | **0.8883** | 305.0 | 45.73 | 2.9125 |
| Qwen2.5-72B-Instruct | 72B | 0.8607 | 205.0 | 48.02 | 4.1985 |
| Llama-4-Scout-17B-16E-Instruct | 109B | 0.8198 | 265.0 | 63.92 | 3.0936 |
| Qwen3-235B-A22B-Instruct-2507 | 235B | 0.8690 | 3640.0 | 13.80 | 0.2387 |
| **POLICYGUARD-4B** | **4B** | 0.8759 | **22.5** | **2.57** | **38.9289** |

Table 5: Efficiency comparison on POLICYGUARDBENCH. We report F1, latency (milliseconds per example), FLOPs per example (converted to TFLOPs), and EA-F1(defined in §4.5). FLOPs for closed-source frontier models are not available ('–'). POLICYGUARD-4B attains competitive F1 with substantially lower latency and compute cost.

## 4.5 INFERENCE EFFICIENCY

While accuracy and F1 capture predictive quality, deployment of guardrails in real-world systems also requires efficiency. To jointly account for predictive performance and inference cost, we report both **FLOPs per example** and a normalized efficiency-aware metric, denoted as **Efficiency-Adjusted F1 (EA-F1)**. Formally, EA-F1 is defined as:

$$\text{EA-F1} = \frac{\text{F1} \cdot L_0}{\text{Latency (ms)}},$$

where *Latency* is the per-example inference latency (in milliseconds), and $L_0$ serves as a baseline latency for normalization. We set $L_0 = 1000$ ms. This formulation captures the trade-off between predictive quality (F1) and processing speed, yielding a measure of efficiency per unit time that is dimensionless and comparable across hardware setups.

Results in Tab. 5 reveal clear trends. Large foundation models such as Llama-3.3-70B and Qwen2.5-72B achieve strong raw F1 scores but latencies of 200–300 ms per example and tens of TFLOPs per input, leading to substantially reduced EA-F1. Frontier models show a similar pattern: while predictive accuracy is competitive, the inference cost remains prohibitive.

By contrast, our fine-tuned POLICYGUARD-4B reaches an EA-F1 of 38.9, more than double the strongest open-source baseline and surpassing frontier models by over an order of magnitude. This reflects a favorable balance: POLICYGUARD-4B delivers competitive predictive accuracy while operating with dramatically reduced latency and FLOPs per example. These results highlight the importance of explicitly optimizing for lightweight policy–trajectory guardrails that are not only accurate but also efficient and practical for real-time deployment.

## 5 CONCLUSION

In this work, we introduced POLICYGUARDBENCH, the first large-scale benchmark for detecting policy–trajectory violations, and proposed POLICYGUARD-4B, a lightweight guardrail model that achieves strong accuracy, cross-domain generalization, and efficient inference. Our experiments demonstrate that safety-oriented guardrails fail to transfer to compliance detection, while policy-specific guardrails can effectively anticipate violations even from early prefixes. These results highlight policy compliance as a distinct and critical dimension of agent reliability, and show that accurate, efficient, and generalizable guardrails are feasible at small scales—laying the foundation for future research on trustworthy policy-compliant agents.

ETHICS STATEMENT

This work does not involve human subjects, personal data, or sensitive information. All agent trajectories in this work are sourced from WebArena, obtained through publicly available links on its leaderboard. Based on these trajectories, we generate and curate policies to ensure diverse coverage while avoiding exposure of private or harmful content. Our contributions focus on developing benchmarks and models for detecting policy compliance in web agents, with the goal of improving the reliability and safety of autonomous systems. We do not release models that could generate harmful or non-compliant behavior; instead, we provide resources for studying detection and mitigation. We are mindful that guardrail technologies can be dual-use, but we believe our emphasis on lightweight compliance detection contributes positively to safer deployment of web agents. We also note that some proprietary models (e.g., Gemini-2.5-Pro) automatically block certain queries, which limited their inclusion in our benchmark.

REPRODUCIBILITY STATEMENT

We make significant efforts to ensure reproducibility. Details of dataset construction, policy synthesis, trajectory standardization, and annotation procedures are described in §3. Comprehensive statistics of POLICYGUARDBENCH are reported in Tab. 1, and train/test splits are explicitly specified to support standardized evaluation. Model architectures, training protocols, and optimization hyperparameters are documented in §3.3 and Appx. §A. All experiments are conducted on H100 GPUs with deterministic decoding settings (temperature set to 0), as noted in §4.1. Results are reported across multiple baselines (Tables 2 to 5). We will release code, dataset, and model checkpoints in an anonymized repository to facilitate verification and future research.

USE OF LARGE LANGUAGE MODELS

We here explicitly document the role of LLMs in our work, LLMs were used in two ways:

First, for data synthesis, LLMs assisted in generating synthetic variations of agent trajectories and candidate policies during dataset construction. All generated outputs were subsequently filtered, validated, and curated by the authors to ensure quality and consistency.

Second, for manuscript preparation, LLMs were used as an assistive tool for polishing the writing, improving clarity, and refining phrasing. The conceptual development, technical contributions, analyses, and final claims of this work are entirely the responsibility of the authors.

No parts of this submission were produced solely by an LLM without human verification. The authors take full responsibility for all content presented in this paper.

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

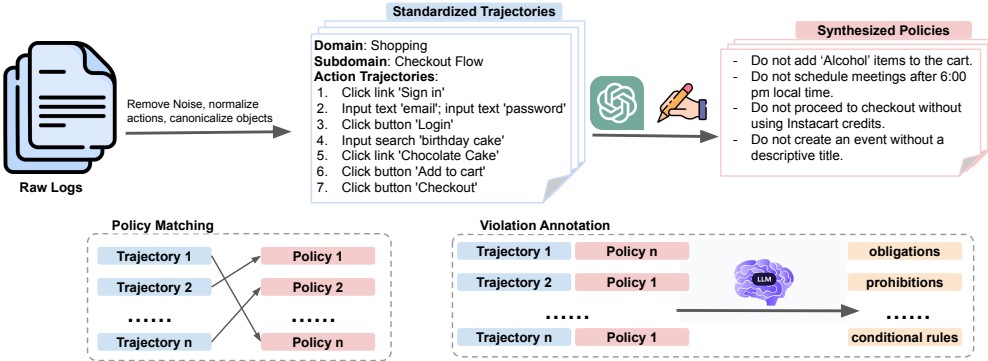

Figure 2: Data processing pipeline for constructing POLICYGUARDBENCH, from raw trajectories to standardized trajectories, synthesized policies, and annotated violations.

## A   OPTIMIZATION DETAILS

We summarize the hyperparameters and optimization setup used in our experiments in Tab. 6.

| Hyperparameter | Value |
|---|---|
| Learning rate | $1 \times 10^{-5}$ |
| Train batch size (per device) | 2 |
| Eval batch size (per device) | 8 |
| Seed | 42 |
| Distributed training | 4 devices |
| Gradient accumulation steps | 8 |
| Total effective train batch size | 64 |
| Total effective eval batch size | 32 |
| Optimizer | AdamW |
| Learning rate scheduler | Cosine |
| Warmup ratio | 0.1 |
| Training epochs | 3 |

Table 6: Hyperparameters and optimization details for model training.

## B   DATA PROCESSING PIPELINE

Fig. 2 illustrates the overall data processing pipeline used in constructing POLICYGUARDBENCH. Starting from raw logs of web agent interactions, we first remove noise, normalize actions, and canonicalize objects to obtain standardized trajectories. From these trajectories, we synthesize diverse policies (covering obligations, prohibitions, and conditional rules) while avoiding sensitive or harmful content. Next, trajectories are matched with corresponding policies, and violation annotations are added to capture whether the trajectory complies with or violates each policy. This pipeline results in a structured dataset of trajectory–policy pairs with compliance labels, which forms the basis for our benchmark.

## C  PREFIX-BASED TRENDS ACROSS MODEL SCALES

To complement the main results, Fig. 3, Fig. 4, and Fig. 5 illustrate prefix-based violation detection accuracy broken down by model scale.

Fig. 3 provides an overview across all evaluated models, showing the general trend that performance is highest at $N = 1$ and decreases as prefixes become longer. To further clarify scale-dependent behavior, we split models into two groups: large-scale and frontier models ($\geq$30B parameters), and smaller open-source models (<30B, including POLICYGUARD-4B).

Large-scale and frontier models, as shown in Fig. 4, start with strong performance at $N = 1$, and although accuracy decreases as more actions are observed, the degradation is relatively moderate. This indicates that larger models can preserve robustness under longer prefixes, albeit at high inference cost.

In contrast, smaller open-source models, as shown in Fig. 5, exhibit greater variance and sharper accuracy drops as $N$ increases. Notably, POLICYGUARD-4B maintains stable accuracy across prefix lengths and remains competitive with much larger systems, underscoring its efficiency–robustness advantage. These trends are consistent with the averages reported in Tab. 3, but reveal more fine-grained behavior across prefix lengths.

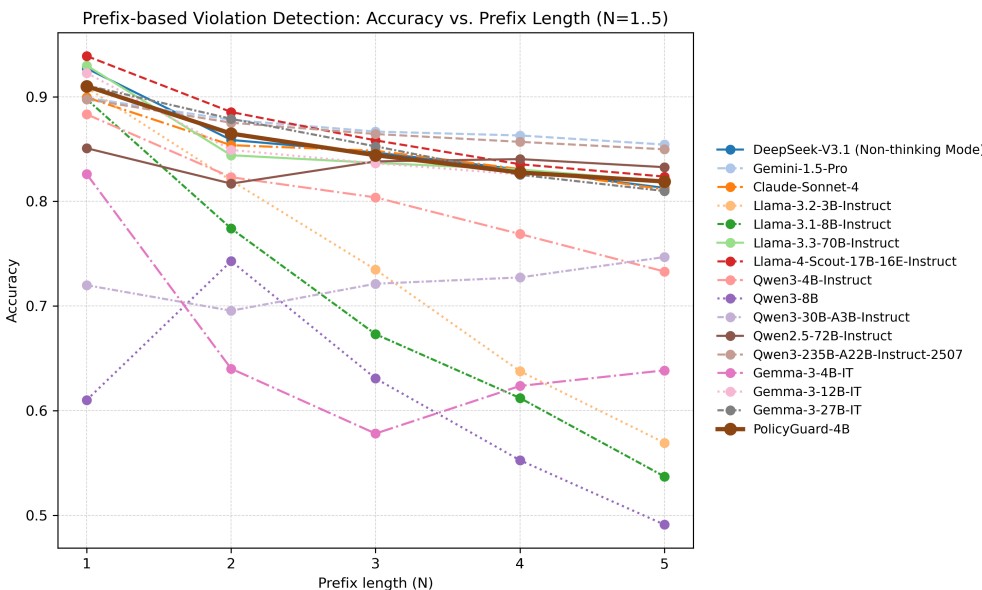

Figure 3: Prefix-based violation detection accuracy across *all evaluated models*. Accuracy is generally highest at $N = 1$ and decreases as prefix length increases.

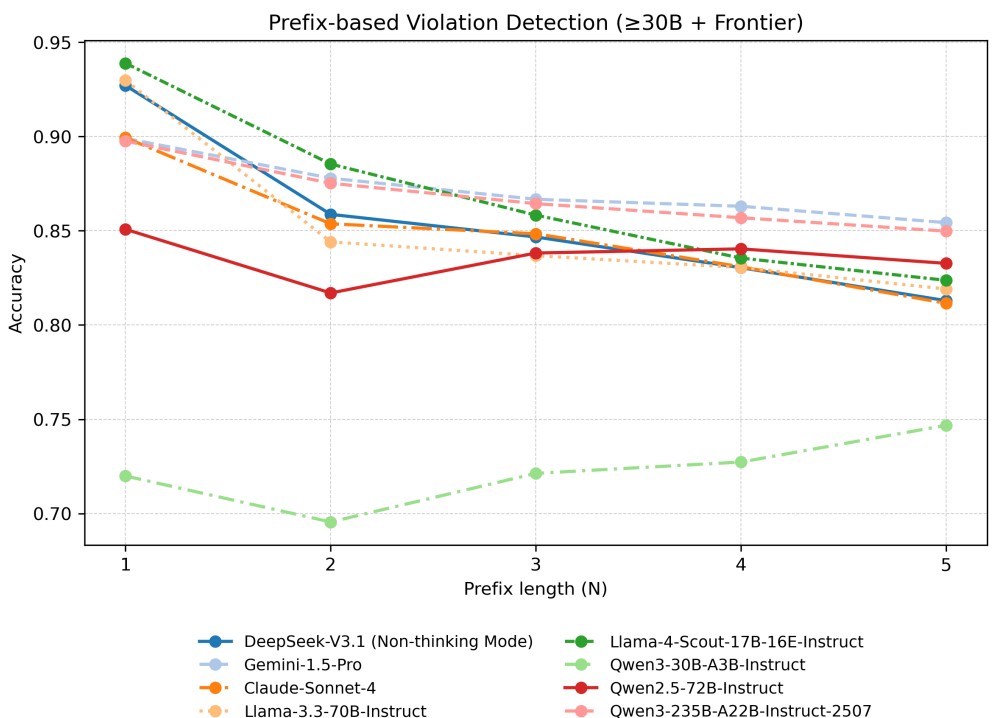

Figure 4: Prefix-based violation detection accuracy for *large-scale models* ($\geq$30B parameters and frontier systems). Performance is strong at $N = 1$ and declines moderately as prefix length increases.

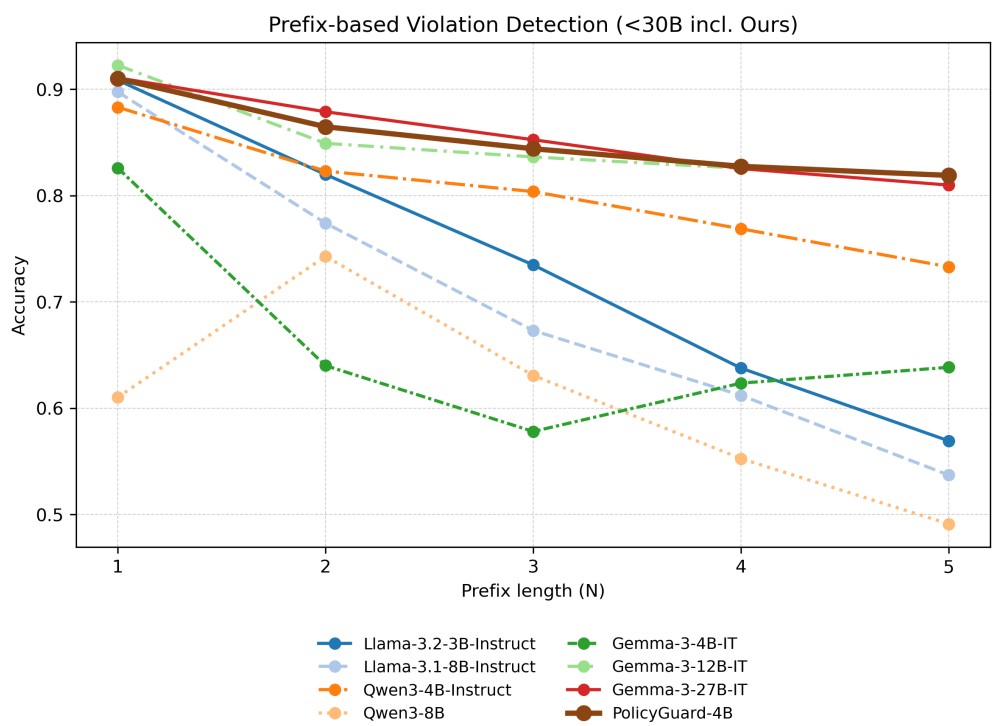

Figure 5: Prefix-based violation detection accuracy for *smaller open-source models* (<30B, including POLICYGUARD-4B). Smaller models show greater variance and sharper drops in accuracy, whereas our lightweight POLICYGUARD-4B achieves robust and competitive results across prefix lengths.

