# OpenReview forum: "Towards Policy-Compliant Agents: Learning Efficient Guardrails for Policy Violation Detection"
_ICLR.cc/2026/Conference — Submitted to ICLR 2026_

### Official Review · Reviewer_onuQ · 2025-10-31

**Soundness:** 3
**Presentation:** 3
**Contribution:** 3
**Rating:** 6
**Confidence:** 3

**Summary:**

This paper introduces POLICYGUARDBENCH, a 60k-scale benchmark for detecting policy violations in web agent trajectories, and proposes POLICYGUARD-4B, a 4B-parameter lightweight guardrail model trained on it. While most existing guardrails target safety (e.g., filtering harmful or insecure actions), the authors argue that policy compliance, adherence to domain- or institution-specific constraints, is an orthogonal dimension of agent reliability. They empirically show that safety-oriented guardrails (e.g., LlamaGuard, ShieldGemma) fail to detect policy violations even when safety risks are absent. To address this gap, the authors construct POLICYGUARDBENCH by systematically generating and annotating policy–trajectory pairs derived from 733 standardized agent trajectories across five domains (Reddit, Map, GitLab, Shopping, Shopping-Admin). Each trajectory–policy pair is labeled as “violation” or “no violation” under formalized rule types (obligation, prohibition, ordering, and conditional). The benchmark also supports prefix-based violation detection, where models must anticipate violations from truncated trajectories, probing early detection capabilities. POLICYGUARD-4B, fine-tuned from Qwen3-4B-Instruct, demonstrates cross-domain generalization and early-prediction accuracy on prefix tasks.

**Strengths:**

The paper distinguishes policy compliance from traditional safety, introducing a new paradigm in agent evaluation. This conceptual clarity fills a gap between safety research and practical deployment constraints in real-world environments.

**Weaknesses:**

1) While the dataset includes 2,195 policies, most are synthetically generated from existing trajectories. The degree of semantic and domain diversity (e.g., regulatory, ethical, or legal policies) is not fully quantified, leaving uncertainty about real-world coverage.

2) The evaluation aggregates all violation types (obligation, prohibition, ordering, conditional) into a single binary metric. Disaggregated performance across these subtypes might provide deeper insight into where the model succeeds or struggles.

3) While POLICYGUARD-4B performs well, its interpretability is limited to prediction-level metrics. A qualitative or attribution-based study could reveal which features (actions, policy clauses) drive violation detection.

**Questions:**

Q1: How consistent are policy–trajectory pairings given the use of embedding-based retrieval and LLM scoring? What proportion of automatically matched pairs were manually corrected during annotation? Could spurious alignments (e.g., unrelated policies matched to trajectories) bias the model toward superficial lexical features?

Q2: GPT-based annotation is calibrated via human seed data. What is the inter-annotator agreement (Cohen’s kappa) between human and model labels? How often did low-confidence labels trigger manual review, and were any systematic biases (e.g., over-flagging prohibitions) observed?

Q3: The prefix-based task reveals highest accuracy at N = 1. Can the authors explain why shorter prefixes outperform longer ones? Does this reflect priors (most early actions are compliant) or detection bias?

---

> ### Author Response · Authors · 2025-11-26
> **Response to Reviewer onuQ**
>
> We thank the reviewer for highlighting the "conceptual clarity" of our work and its value in bridging the gap between safety research and practical deployment. We address your specific questions below.
>
> ### 1. Pairing Consistency & Spurious Alignments
> > How consistent are pairings? Could spurious alignments bias the model toward superficial lexical features?
>
> 1. Preventing Irrelevant Matches: We explicitly mitigate spurious alignments (matches that share keywords but are unrelated) through our two-stage pipeline. While embedding retrieval ensures we find enough candidates, the subsequent LLM-based scoring enforces strict logic, filtering out pairs that lack a real semantic dependency.
> 2. Generalization beyond Keywords: The model's ability to reason beyond simple keywords is empirically proven by the Out-of-Domain (OOD) experiments in Table 4. If the model relied on simple lexical matching from the training set, it would fail to generalize to unseen domains where the vocabulary differs significantly. The stable OOD performance confirms it learns the underlying compliance logic.
>
> ### 2. Annotation Agreement
> > What is the inter-annotator agreement?
>
> Please refer to _General Response Section 1_. Our human-verified reference set (N=500) achieved over 90% agreement with the automated labels. This high consistency suggests that systematic biases are minimal and the definitions of violations are objective.
>
> ### 3. Prefix-based Task Accuracy
> > Why does N=1 reveal the highest accuracy?
>
> We attribute this to the structural nature of the task:
> 1. Highly Structured Early Actions: Through our observation of the data, early steps in WebArena tasks are standardized. Violations at this stage are binary and unambiguous.
>
> 2. Ambiguity in Length: Longer prefixes (e.g., N=5) introduce significantly more branching and state accumulation. This increases the ambiguity of whether a long-horizon constraint is definitively violated or still recoverable, making prediction harder compared to the "clean slate" of N=1.

---

### Official Review · Reviewer_Z8ni · 2025-11-01

**Soundness:** 3
**Presentation:** 2
**Contribution:** 2
**Rating:** 4
**Confidence:** 2

**Summary:**

The paper proposes a new benchmark and model framework for ensuring policy compliance in autonomous web agents.
The results show the proposed 4B model achieves 90.1% accuracy and 87.6% F1, outperforming much larger models (e.g., Llama-3.3-70B, Qwen2.5-72B) and closed-source systems (Gemini, Claude) while maintaining low inference latency (22.5 ms/example). The model generalizes well across unseen domains and performs robustly under prefix-based (early detection) settings.

**Strengths:**

1. Novel focus on policy compliance.
This paper identifies a previously underexplored dimension of agent reliability: compliance with human or institutional policies, distinct from safety or task success. This conceptual contribution broadens the landscape of responsible AI and agentic system evaluation.
2.The new benchmark is appreciated.
The dataset construction is meticulous—covering multiple domains, systematic policy synthesis using LLMs, and dual within- and cross-subdomain evaluations. The inclusion of prefix-based tasks allows studying early detection of violations, which is both original and practically relevant

**Weaknesses:**

1. Most data are generated and not validated by human.
2. The coverage is limited in terms of context.

My major concern of this work is the generalization and how practical this work is.  Only a few domains are covered and I think it is better to include more. Additonally, I'm a little bit worried about the quality of the benchmark. I appreciate the efforts but there should be more detailed validation and analysis on the data.

For example, for the trajectories, maybe it is better to formulate the "policy" to formal specifications and then the violation can be checked by monitoring. For example, linear temporal or first-order logic. These monitoring and checks can provide rich feedback and formal guarantee on the results and label and I think this is missing in current version.

**Questions:**

N/A

---

> ### Author Response · Authors · 2025-11-26
> **Response to Reviewer Z8ni**
>
> We thank the reviewer for their thoughtful feedback. We address your specific concerns regarding data quality, coverage, and formal verification below.
>
> ### 1. Data Quality and Human Validation
> > Most data are generated and not validated by human.
>
> We respectfully clarify that validation is a core part of our pipeline. As detailed in _General Response Section 1_, we conducted a rigorous manual verification on a representative subset of 500 trajectory-policy pairs, achieving over 90% agreement with our automated pipeline. This confirms the data quality is high. Furthermore, we are committed to expanding this human verification effort to cover a larger portion of the dataset in the final revision to further guarantee robustness.
>
>
> ### 2. Domain Coverage and Generalization
> > Coverage is limited... only a few domains are covered.
>
> We address the concern about limited coverage through rigorous generalization testing. Beyond the standard domains, we conducted extensive **Out-of-Domain (OOD)** experiments using a "Leave-One-Domain-Out" protocol, as detailed in **Section 4.4 of the paper** (Table 4) and _General Response Section 2_. These experiments demonstrate that our model can effectively generalize to completely unseen environments (e.g., training on Reddit/GitLab/Map and testing on Shopping), proving that it learns robust compliance reasoning rather than overfitting to specific domain artifacts.
>
> ### 3. Formal Specifications
> > Formulate the 'policy' to formal specifications... e.g., linear temporal logic.
>
> We agree that formal methods offer rigorous guarantees, but we chose a Natural Language approach for specific reasons relevant to autonomous agents:
>
> 1. Real-world Applicability: In real-world deployments, policies often come from unstructured user instructions rather than pre-written Linear Temporal Logic formulas. Mapping arbitrary web DOM states to rigid logic symbols is brittle and hard to scale across changing websites.
>
> 2. Schema as "Soft" Formalization: To balance flexibility with rigor, we implemented Schema Normalization, as mentioned in **General Response Section 1**. By mapping policies to structured categories, we capture the logical essence needed for monitoring without losing the semantic flexibility of LLMs.

---

### Official Review · Reviewer_4BD3 · 2025-11-03

**Soundness:** 2
**Presentation:** 2
**Contribution:** 3
**Rating:** 4
**Confidence:** 2

**Summary:**

A challenge in practice is policy violation detection in long-horizon trajectories that are collected under externally imposed or human-defined policies in autonomous web agents, especially in the settings of multiple contexts and subdomains. This work introduces a benchmark called POLICYGUARDBENCH to provide 60k examples for this policy violation detection task, by providing both within subdomain and cross subdomain pairings with violation labels and policies. The raw data used in this work is from an existing benchmark SCRIBEAGENT. A benchmark algorithm called POLICYGUARD-4B is also introduced in this benchmark testbed.

**Strengths:**

- From my perspective, this paper investigates an interesting practical challenge, policy violation detection, that can broadly exist in real-world scenarios but may be neglected by many policy learning approaches.
- The introduced benchmark should be interesting and beneficial to related domains such as RL, safety, and system designs, etc.
- The introduced algorithm POLICYGUARD-4B shows superior performance in terms of latency and EA-F1 in efficiency.
- The introduced benchmark is examined via multiple evaluation metrics and case-studies.

**Weaknesses:**

- The motivation of introduced algorithm is not very clear to me. Why POLICYGUARD-4B is in need and what guides the specific design of POLICYGUARD-4B? I assume latency is a part of motivation, but I would prefer to see a discussion from the authors.
- Some details of benchmark design seem missing. Please refer to my questions in details.

**Questions:**

Besides the weaknesses listed above, regarding benchmark design, I was also wondering:
- The details of candidate policy generation seem missing. Could you provide further details on how you trained your policies and how you choose which algorithms (as you only synthesize 2-3 policies in total)?
- Do results in Table 2 averaged across all subdomains?

---

> ### Author Response · Authors · 2025-11-24
> **Response to Reviewer 4BD3**
>
> We thank the reviewer for their constructive feedback and for identifying areas where our motivation and benchmark details needed further clarification. We address your specific questions below.
>
> ### 1. Motivation and Design of POLICYGUARD-4B
> > *"Why POLICYGUARD-4B is in need and what guides the specific design... I assume latency is a part of motivation..."*
>
> We agree that latency is a primary motivator, but the need for POLICYGUARD-4B stems from two critical gaps in the current landscape:
>
> - **Gap 1: The Orthogonality of Safety vs. Compliance.** As detailed in _General Response Section 3_, current safety models (like LlamaGuard) often fail here because they only look for harmful content rather than specific rule violations. POLICYGUARD-4B is designed to specifically bridge this accuracy gap where generic safety models fail.
> - **Gap 2: The Efficiency-Accuracy Trade-off.** While the frontier models (e.g., Gemini-1.5-Pro, Claude-Sonnet-4) are accurate, they are prohibitively slow and costly for real-time "gatekeeping." Conversely, small open-source models often lack the reasoning capability for this task.
> * Our Solution: Table 2 and Table 5 demonstrate that POLICYGUARD-4B effectively fills this void. It achieves accuracy competitive with 70B+ models (90.1%) while maintaining the low latency (22.5ms) required for practical deployment, outperforming baselines significantly on the Efficiency-Adjusted F1 metric.
>
> ### 2. Details on Candidate Policy Generation
> > *"The details of candidate policy generation seem missing. Could you provide further details on how you trained your policies and how you choose which algorithms (as you only synthesize 2-3 policies in total)?"*
>
> There appears to be a slight misunderstanding regarding the policy count. To clarify: we synthesize 2-3 policies **per trajectory**, not in total. Our final dataset contains 2,195 unique curated policies matched across thousands of trajectories.
>
> Regarding the specific algorithms for generation and matching, please refer to _General Response Section 1_, where we provide a detailed breakdown of the synthesis pipeline and quality assurance measures.
>
> ### 3. Clarification on Table 2 Results
> > *"Do results in Table 2 averaged across all subdomains?"*
>
> **Yes.** The results reported in Table 2 represent the global average across the entire test set. This includes all 5 domains (Shopping, Shopping Admin, Reddit, GitLab, Map) and aggregates performance across both "Source" and "Target" generalization splits to provide a holistic view of model performance. Please refer to the explanation in _General Response Section 2_ for specifics.

---

### Official Review · Reviewer_fAdg · 2025-11-04

**Soundness:** 2
**Presentation:** 2
**Contribution:** 2
**Rating:** 2
**Confidence:** 3

**Summary:**

This paper introduced a benchmark dataset claimed to be facilitating trajectory-level policy compliance, which started with raw trajectories from ScribeAgent and WebArena followed by data standardization (i.e., cleaning and assigning domains and sub-domains), synthesizing policies (i.e., getting the policies that the agent needed to follow) and labeling (i.e., trajectory-policy compliance check). Authors also fine-tuned a specific version of baseline LLM (Qwen3-4B-Instruct) as an additional baseline to the benchmark dataset. The benchmark itself not only introduced regular hold-out type binary classification tasks (i.e., complied or not), but prefix-based violation detection (i.e., when less information are available), and cross-domain generalization.

**Strengths:**

* The motivation of setting up this benchmark is clear and the reviewer has not found other peer-reviewed and published benchmark that specifically focuses on trajectory-level policy compliance.
  * Note that there exists some highly related non-benchmark works (e.g., LlamaGuard 3 and 4) that were based off from a very similar setup, i.e., to classify if a series of questions and answers (i.e., essentially trajectories as considered in the paper) violates the policy/guidelines or not. And the authors found that LlamaGuard did not perform as well in this benchmark (which has multiple policies) than in the original setup (which only considered a single policy).
* The benchmark provided 2 additional setups beyond the typical hold-out train/test split, i.e., prefix-based violation detection and domain generalization.

**Weaknesses:**

* There are quite a few strong statements the authors made in the paper (mostly in the introduction and related work/background sections) that were left to be justified.
 > around lines 118, *These efforts are well motivated, since safety is a prerequisite for deployment, but they often result in
over-defensiveness: models may over-refuse benign behaviors in order to minimize risks, thereby
reducing usability in practice.*

    * Is this a conclusion drawn by the authors following their benchmark, or a common ground that safety-oriented works including LlamaGuard has been widely recognized as often being *overly defensive* that limited their usability in practice?

  > around lines 126, *Notably, some prior works have treated policy violations as equivalent to unsafe behaviors (Chen et al., 2025a;b; Vijayvargiya et al., 2025), an assumption that we argue is unfounded. Our own tests further corroborate this view: safety-oriented guardrails such as LLAMAGUARD3 (Meta Llama Team, 2024) and LLAMAGUARD4 (Meta Llama Team, 2025) often classify trajectories that clearly violate policies as safe.*

    * The reviewer did not find any other supporting evidence or justification for the argument presented in the beginning of the quote above, i.e., the latter half of the quote itself arguable did not sufficiently justify. Specifically, the authors did show that under their benchmark setup, LlamaGuard did not perform as well as other baselines. However, could the deciding factor be that LlamaGuard only considered a single policy to comply with rather than multiple given their use case? Moreover, even one assumes that there are facts to support that the safety-oriented guardrail works do often violate policies, how's that connected to the statement that *treating  violations as equivalent to unsafe behaviors* was deemed unfounded by the authors? Besides, are there any other references implying or supporting this statement?

* Following from the 2nd quote above, it was also unclear to the reviewer what were the formal definitions of *policy violations* and *unsafe behaviors*. Were the authors implying that one could be a subset of another, or there could be parts under each definition that are mutually exclusive? If the latter, how this benchmark reflected such nuances?

* There are also unclarity in some of the design choices in data processing/policy synthesis.
  > lines 170-172, *To transform these into a clean and uniform representation, we remove noise (e.g., empty or duplicate events, rendering artifacts), normalize actions into a controlled verb set (Click, Input, Scroll, Select, Navigate, Submit, etc.), and canonicalize objects (e.g., link ’My Account’ or button ’Search’). Redundant operations are merged, and excessively long trajectories may be truncated. Each trajectory is then assigned a domain and subdomain, which are propagated to all steps.*

  * What were the raw actions and how were them being grouped into the controlled verb set? What was the coverage after grouping? What defines the redundant operations and how were they merged? How long a trajectory would be considered *excessively long* and how the truncation was done?
    * Beyond these clarifications, the reviewer would also appreciate some stats showing the histogram on how often each element in the controlled verb set appeared.

  > around lines 174, *Each trajectory is then assigned a domain and subdomain, which are propagated to all steps.*

  * What were the criteria for labeling domains and sub-domains. For example, would a reddit thread on sharing cloth purchases opinions considered *Reddit* or *Shopping*? From Table 1 it looked like authors categorized all trajectories into five domains, i.e., reddit, map, gitlab, shopping_admin, shopping. How did the authors determine to classify into these five specific categories and what were the criteria used in the categorization? What was the difference between *shopping* and *shopping_admin*?

  > lines 186, *Candidate policies are first synthesized by GPT-4o1 and subsequently curated to ensure quality and consistency.*
  * The reviewer was wondering how to check the validity of these policies, as 2-3 were generated by GPT for every single trajectory where all needed to be human-reviewed. Are there any criteria to determine which GPT-generated policies needed to be modified, and in what ways? For example, the policies have to make sense but more importantly, do they have to reach some degree of complexity?
    * The authors also claimed the following as one of the challenges in studying trajectory-level compliance around lines 52, *First, policies are inherently diverse and may originate from human instructions, institutional rules, or environmental constraints, which makes it non-trivial to align them with equally diverse trajectories.*
    * The reviewer was wondering if the human-tuned on top of GPT-generated policies would naturally reflect these various types of constraints, i.e., the reviewer could see some human instructions being ported in during the manual correction process, how did the GPT was setup so the policies capture the institutional rules, environmental constraints, etc.?

  > around lines 195, *First, candidate policies are retrieved for each trajectory using embedding-based similarity (Reimers & Gurevych, 2019) and keyword triggers (e.g., element names such as confirm or delete). These candidates are further refined using heuristic rules and LLM-based scoring to ensure that the policies are semantically relevant and checkable.*

  * The reviewer was curious about to what extend the embedding-based similarity would ensure proper matching and what were the criteria used to determine if further refinement is needed or not. If so, what were the heuristic rules used for refinement. Moreover, how reliable the LLM-based scoring is to quantify the relevancy between the trajectory and policy? Are there any stats/analyses that could reflect these?

  > around lines 209, *Each trajectory–policy pair is labeled as either violation or no violation. We define operational criteria covering common rule types: obligations (a required action is missing), prohibitions (a forbidden action is present), ordering constraints (steps occur in the wrong sequence), and conditional rules (a consequent is not satisfied when the antecedent holds).*

  * The reviewer was curious how these 4 types of violations distribution across the train/test sets, and what was the thinking behind converting the task to a binary *violation/no_violation* one instead of a multi-class tasks pinpointing the type(s) of violations?

  > around lines 249, *for violation cases, we truncate trajectories to the first N steps (N = 1. . . 5), re-match them with their corresponding policies, and re-label the resulting pairs.*

  * If the purpose of the prefix-based splits was to study how the agent would perform with incomplete observations, the reviewer did not get why policy re-matching was needed. Couldn't the policy stays the same while just the trajectory got truncated?

* The 3rd contribution of the paper seemed to be over claimed to some extent, i.e., as stated by the authors *We develop POLICYGUARD-4B, a lightweight and effective guardrail model that attains strong accuracy, cross-domain generalization, and state-of-the-art efficiency, demonstrating the practicality of small-scale guardrails in real-world deployment*. PolicyGuard-4B indeed performed better on top of its backbone after fine-tuning with the dataset curated by the authors. However, it does not facilitate an apple-to-apple comparison against other baseline models that are not fine-tuned. To the reviewer's understanding that the improvement was mostly attributed to the use of the benchmark dataset during fine-tuning, so it was not clear why a single model fine-tuned on the dataset could contribute? In contrast, the contribution brought in by the benchmark dataset could be more thoroughly justified by showing improvements for most baselines after fine-tuning.

* Moreover, the reviewer thought it would be equally important to re-test models like LlamaGuard on their original tasks after fine-tuning with the benchmark dataset, ensuring that their performance on the original tasks were not significantly degraded. Otherwise, the performance gain from fine-tuning could mainly be attributed to data/topic biases, i.e., only train the models to do some tasks better than another. In the reviewer's opinion, this would also be the key showing the generalization ability of the agent, part of which was missing in the paper.

**Questions:**

See the section above

---

> ### Author Response · Authors · 2025-11-25
> **Response to Reviewer fAdg 1**
>
> We appreciate the reviewer for their thoughtful comments and for identifying the need to better justify our motivation regarding "over-defensiveness" and the distinction between safety and compliance. These are critical aspects of our work, and we address your specific questions below.
>
> ### 1. Over-defensiveness
> > Is the statement about "over-defensiveness" a common ground?
>
> This is a widely recognized consensus in recent literature, formally described as the Safety Alignment Tax—the phenomenon where aligning models for safety leads to a degradation in general utility, specifically manifested as high False Refusal Rates on benign instructions.
>
> We support this with three recent pivotal studies:
>
> [1] [Bianchi et al. (ICLR 2024)](https://arxiv.org/pdf/2309.07875): Systematically demonstrated that standard safety tuning leads to exaggerated safety, causing models to fail at following benign instructions due to over-sensitivity.
>
> [2] [Cui et al. (ICML 2025)](https://openreview.net/pdf?id=CdFnEu0JZV): Quantified this tax using OR-Bench, revealing that leading models exhibit significant over-refusal on harmless queries.
>
> [3] [Röttger et al. (NAACL 2024)](https://arxiv.org/pdf/2308.01263): Introduced XSTest, showing that models frequently refuse safe prompts (e.g., "How to kill a Python process") solely due to lexical triggers, confirming that generic safety guardrails are often too defensive for precise policy compliance.
>
> ### 2. Violations vs. unsafe
> > Could LlamaGuard's failure be due to considering only a single policy? And how does this connect to your argument that "treating violations as unsafe is unfounded"?
>
> 1. LlamaGuard failed despite having the correct policy.
> We clarify that LlamaGuard was explicitly provided with the relevant policies. Its failure is due to a semantic blind spot: it is trained to detect inherent harm. Accordingly, when it encounters a benign action that violates a logical rule, it predicts "Safe" because there is no harm, confirming that generic safety models do not generalize to logical compliance.
>
> 2. Why "Violation = Unsafe" is unfounded.
> Our justification rests on the fundamental distinction between inherent harm and contextual correctness. Safety guardrails are designed to block universally harmful content , whereas policy compliance enforces state-dependent rules on otherwise benign actions. For example, "spending 205 dollars" is inherently safe but becomes a violation only when conditioned on a "$200 limit" policy. Generic safety models consistently fail to capture this distinction. Please refer to _General Response Section 3_ ("Safety vs. Compliance") for our detailed framework.

---

> ### Author Response · Authors · 2025-11-25
> **Response to Reviewer fAdg 2**
>
> We thank the reviewer for their detailed and rigorous questions regarding our data pipeline and experimental setup. We address your specific points below.
>
> ### 1. Data Processing and Action Standardization (Lines 170-172)
> > What were the raw actions? How were they grouped? What defines redundant operations? How was truncation done?
>
> * Raw Actions & Grouping: Raw logs record heterogeneous browser events, like clicks or inputs. We mapped these into a controlled verb set (Click, Input, Scroll, Select, Navigate, Submit) to standardize the action space.
> * Redundancy: We defined redundant operations as consecutive actions that do not trigger a DOM state change (e.g., duplicate clicks, navigation loops). These were merged to ensure trajectory efficiency.
> * Truncation: "Excessively long" refers to trajectories exceeding context limits 4096 tokens. We enforced a strict maximum length of 15 steps to maintain manageable reasoning contexts while preserving final outcomes.
>
> ### 2. Domain and Sub-domain Definitions (Line 174)
> > Criteria for labeling domains? Difference between Shopping and Shopping_Admin?
>
> * Definitions: Please refer to _General Response Section 2_. Domains are adopted directly from WebArena, while sub-domains are derived from action semantics.
>
> * Shopping vs. Admin: These are distinct environments defined by WebArena. "Shopping" is the customer frontend, while "Shopping Admin" is the backend CMS; they possess completely different action spaces and UI layouts.
>
>
> ### 3. Policy Validity & Matching (Lines 186, 195)
> > Validity of GPT-generated policies? Reliability of embedding matching?
>
> Please refer to _General Response Section 1_. We detail our hybrid synthesis pipeline, the specific use of `all-MiniLM-L6-v2` for retrieval, and the human-verified reference set (N=500) that achieved >90% agreement, ensuring policies are valid and properly matched.
>
> ### 4. Violation Types & Binary Classification (Lines 209)
> > Distribution of 4 violation types? Why binary instead of multi-class?
>
> - Rationale: Please refer to _General Response Section 3_. We prioritize binary classification to serve the practical "Gatekeeping" function of a guardrail.
> - Distribution: While we focus on binary detection, our dataset annotations cover all four types. We will include the specific histogram of these types in the final Appendix to support fine-grained analysis.
>
> ### 5. Prefix-based Splits & Re-matching (Line 249)
> > Why was policy re-matching needed? Couldn't the policy stay the same?
>
> - Logical Necessity: Violations are state-dependent. A trajectory violating a "Budget < $200" policy at Step 10 (205 dollars total) is compliant at Step 5 (50 dollars total).
> - Preventing Hallucination: Re-matching ensures the ground truth reflects the *current* partial state. Keeping the original label would force the model to hallucinate a future violation that has not yet occurred.
>
> ### 6. Model Contribution & Apple-to-Apple Comparison
> > Is the contribution mostly the dataset? Why not fine-tune baselines?
>
> We agree the dataset is a foundational contribution, but PolicyGuard-4B serves as a proof-of-concept for **Efficiency**.
>
> - Feasibility: Our goal was to show that a small model (4B) can master this complex reasoning task. Comparing it against non-fine-tuned frontier models highlights that specialized data + small compute > generic data + massive compute for this specific application.
>
> - Baseline Improvements: We acknowledge your point. In the revised version, we will include results of fine-tuning other open baselines (like LLamaGuard, ShieldGemma series) on our dataset to separate the data contribution from the model architecture contribution.
>
> ### 7. LlamaGuard & Catastrophic Forgetting
> > Important to re-test models on their original tasks.
>
> We appreciate this insight. However, we view PolicyGuard as a specialized module intended to run alongside general safety guardrails in a mixture-of-experts setup, **rather than replacing them**.
>
> As argued in _General Response Section 3_, Safety and Compliance are orthogonal. A PolicyGuard model should focus on business policy logic, while LlamaGuard handles toxicity. We also advocate for a modular deployment where LlamaGuard handles toxicity and PolicyGuard handles business logic, avoiding the trade-off of catastrophic forgetting.

---

### Author Response · Authors · 2025-11-24
**General Response**

We sincerely thank all reviewers for their time and the constructive feedback dedicated to our work. We are encouraged that the reviewers recognized our problem setting as "novel and important" and found our benchmark to be "comprehensive."

In this general response, we address three common themes raised across the reviews to clarify our contributions and design choices: (1) The reliability of our data synthesis and human evaluation pipeline; (2) The definitions of domains versus subdomains and our generalization settings; and (3) The rationale behind our task formulation.

### 1. Policy Synthetic Data & Human Evaluation Pipeline
Reviewers inquired about the validity of GPT-generated policies and the reliability of the trajectory-policy matching process.

1.1 Implementation Details on Matching: We utilized `all-MiniLM-L6-v2`[1] to encode both trajectories and policies. This lightweight model was chosen for its efficiency in semantic search. We found that this embedding retrieval, followed by a keyword trigger filter, effectively recalled relevant candidate policies.

1.2 Quality Assurance Pipeline: We enforced data quality by mapping raw policies to strict schemas (Obligations, Prohibitions, Conditional Rules)  and validating the pipeline against a human-verified subset of 500 randomly sampled pairs. This reference set achieved over 90% agreement with our automated annotations, confirming that our dataset reflects valid, objective constraints rather than hallucinations.

### 2. Domain, Sub-domain, and Cross-Domain Generalization

2.1 Definitions:
- Domains: Our domain definitions (e.g., Shopping, GitLab, Reddit) are directly adopted from the WebArena environment [2]. We constructed our benchmark using the public trajectories generated by ScribeAgent [3], strictly preserving the original domain metadata found in these logs. This ensures our categorization is fully consistent with standard agent evaluation frameworks.

- Sub-domains: Since a single domain like Shopping contains diverse activities, we derived subdomains by analyzing the semantic information within the trajectory's action sequence, like URLs and button clicks.

2.2 Generalization Logic:
- Source (Within-Distribution): These pairs consist of policies and trajectories drawn from the same subdomain. This tests the agent's ability to detect violations in familiar contexts.

- Target (Out-of-Distribution): To test robustness, we pair policies from one subdomain with trajectories from a different subdomain within the same domain. This corresponds to the "Target" statistics in Table 1. This setting mimics real-world scenarios where a global rule must be enforced across varying page layouts and interaction flows.

### 3. Safety vs. Policy Compliance
We reiterate that Policy Compliance is orthogonal to Safety, and treating "violation = unsafe" is fundamentally problematic. Based on our framework, we distinguish these concepts as follows:

- Unsafe Behavior: Refers to actions that cause inherent harm, violate universal safety principles, or result in security/privacy failures.
- Policy Violation: Refers to the failure to satisfy a task-specific, user-specific, or environment-specific rule, regardless of harmfulness.

Our experiments demonstrate that existing safety guardrails (like LlamaGuard) fail to capture these nuances because they focus on inherent harm rather than state-dependent constraints. The relationship between these concepts is illustrated in Table A below:

Table A: The Relationship between Policy Violations and Unsafe Behaviors

| Case                                     | Example                       |
|------------------------------------------|---------------------------------|
| Intersection exists (unsafe ∩ violation) | Accessing private user data    |
| Violation but benign                     | Exceeding budget constraints     |
| Unsafe but no explicit violation         | Hallucinating a sensitive email  |


**Rationale for Binary Classification:**
Given this distinction, we formulated the task as Binary Classification for practical deployment. In real-world systems, the primary function of a guardrail is "Gatekeeping"—deciding whether to block an action. While we annotate types for analysis, the binary signal is the critical operational requirement for preventing the diverse range of violations shown above.

[1] [Augmented SBERT: Data Augmentation Method for Improving Bi-Encoders for Pairwise Sentence Scoring Tasks](https://aclanthology.org/2021.naacl-main.28/) (Thakur et al., NAACL 2021)

[2] [WebArena: A Realistic Web Environment for Building Autonomous Agents](https://arxiv.org/pdf/2307.13854) (Zhou et al., ICLR 2024)

[3] [ScribeAgent: Towards Specialized Web Agents Using Production-Scale Workflow Data](https://arxiv.org/abs/2411.15004) (Shen et al., arxiv 2024)

---

### Meta-Review · Area_Chair_E57d · 2026-01-07

**Summary:**

The paper argues about the challenge of making sure that autonomous web agents adhere to specific, context-dependent policies (e.g., budget constraints or prohibitions on certain items). The paper provides a synthetic dataset PolicyGuardBench of trajectory-policy pairs derived from WebArena logs with policies synthesized by GPT-4. Additionally, a lightweight finetuned model PolicyGuard-4B is also provided.

Overall, I think the problem setup is a bit contrived and not justified properly. The main concern which was mentioned by Reviewers Z8ni, fAdg, and onuQ, is the full reliance on LLMs for policy synthesis, matching, and annotation. I am not sure if another synthetic dataset would provide useful signal for research. The community generally would benefit from a higher degree of human validation. Multiple reviewers also found the justification of PolicyGuard-4B and its comparison only with zero-shot/prompted models a bit weak.

I recommend rejecting the paper and request the authors to consider comments by the reviewers for future submission.

**Reviewer Concerns:**

The additional details on the data processing pipeline helped clarify the mechanics of dataset construction but the core issues remain outstanding.

**Reviewer Scores:**

Other than Reviewer fAdg, no one other reviewer would have increased their score.

---

### Decision · Program_Chairs · 2026-01-26

Reject